# Penisimplicins A and B: Novel Polyketide–Peptide Hybrid Alkaloids from the Fungus *Penicillium simplicissimum* JXCC5

**DOI:** 10.3390/molecules29030613

**Published:** 2024-01-27

**Authors:** Qing-Yuan Wang, Yang Gao, Jian-Neng Yao, Li Zhou, He-Ping Chen, Ji-Kai Liu

**Affiliations:** 1School of Chemistry and Materials Science, South-Central Minzu University, Wuhan 430074, China; 2School of Pharmaceutical Sciences, South-Central Minzu University, Wuhan 430074, China; 3Yunnan Key Laboratory of Pharmacology for Natural Products & School of Pharmaceutical Science, Kunming Medical University, Kunming 650500, China; yaojianneng@kmmu.edu.cn

**Keywords:** *Penicillium simplicissimum*, polyketide, nitrogen-containing compounds, structural elucidation, computational chemistry, acetylcholinesterase inhibition

## Abstract

In this study, two previously undescribed nitrogen-containing compounds, penisimplicins A (**1**) and B (**2**), were isolated from *Penicillium simplicissimum* JXCC5. The structures of **1** and **2** were elucidated on the basis of comprehensive spectroscopic data analysis, including 1D and 2D NMR and HRESIMS data. The absolute configuration of **2** was determined by Marfey’s method, ECD calculation, and DP4+ analysis. Both structures of **1** and **2** feature an unprecedented manner of amino acid-derivatives attaching to a polyketide moiety by C-C bond. The postulated biosynthetic pathways for **1** and **2** were discussed. Additionally, compound 1 exhibited significant acetylcholinesterase inhibitory activity, with IC_50_ values of 6.35 μM.

## 1. Introduction

Alzheimer’s disease (AD) is a central nervous system degenerative disease which results in the progressive and irreversible loss of brain function and resultant behavioral changes. Since the first case of Alzheimer’s disease was discovered in 1906, nearly 50 million people have been diagnosed with AD [1,2]. Nowadays, AD has been recognized as a global public health priority by the World Health Organization (WHO) [3]. One of the well-established theories on the causes of AD suggests that the neurotransmitter acetylcholine levels are too low in the brains of AD patients. Current treatment approaches for this disease are mainly based on the cholinergic hypothesis and specifically on the cholinergic inhibition. Acetylcholinesterase (AChE) inhibitors can significantly alleviate the symptoms of AD, and they are the most effective treatments at present [2]. Drugs such as donepezil [4], rivastigmine [5], and galantamine [6] have been approved as AchE inhibitors for regular agencies by the US Food and Drug Administration (FDA) and the European Medicines Agency (EMA) [2]. However, the concern is that these drugs have side effects such as diarrhea, nausea, vomiting, and so on [7]. Moreover, they are only effective for patients with mild-to-moderate AD [8], and sometimes have a different therapeutic effect in treating different phenotypes of apolipoprotein E-genotyped patients, with the apolipoprotein E gene a well-known gene that influences Alzheimer’s risk [9].

Natural products have played, and are still playing, important roles in drug research and discovery. Thus, it is of great importance to explore new efficient and low-toxicity AChE inhibitors from natural sources. Recently, many natural products have been reported to exhibit anti-acetylcholinesterase activity, such as floribundiquinone B (IC_50_ = 5.95 μg/mL) from the roots of *Berchemia floribunda* [10], biatractylolide (IC_50_ = 6.5458 μg/mL) from *Atractylodis macrocephalae* Rhizoma [11], and isoimperatorin (IC_50_ = 23.1 μM) and 6′-hydroxy-7′-methoxybergamottin (IC_50_ = 13.2 μM) from the fruit peels of *Citrus hystrix* [12]. However, most of these products were isolated from precious and limited plant sources.

The genus of *Penicillium* has proved to be a prolific source of natural products. The secondary metabolites from this genus contain plenty of structural types, such as nitrogen compounds [13], terpenoids [14], polyketides [15], etc. Many of these have been successfully developed as clinically used drugs, such as mycophenolic acid and compactin [16,17]. Recently, Dai et al. reported a series of quinolone alkaloids from *Penicillium simplicissimum* with inhibitory activity on NO production [18,19]. This study concentrated on the secondary metabolites of the fungus *Penicillium simplicissimum* JXCC5, which was isolated from the rhizosphere soil of the insect pathogenic fungus *Ophiocordyceps sinensis* collected in wild forest. Some interesting molecular weights of nitrogen-containing compounds were found in the LC-MS data analysis of the crude extract in the early stages of their study (Appendix A). Extensive follow-up endeavors achieved two novel nitrogen-containing compounds, penisimplicins A (**1**) and B (**2**), in isolation. These two compounds both exhibited a unique skeleton featured by an amino acid-derivative linked to a polyketide moiety by C-C bond. Fungal polyketide–peptide hybrids (PK-NRP), which are manufactured by polyketide synthase–nonribosomal peptide synthetase (PKS-NRPS), account for a large group of biologically active and structurally intriguing natural products. Canonical fungal PKS-NRPS usually uses the PKS module and NRPS module to assembly the hybrid products. The NRPS module always contains a condensation domain (C domain) to yield amino acylated adduct. By this point, compounds **1** and **2** are also classified into PK-NRP but are proposed to be biosynthesized by noncanonical biosynthetic pathways. Herein, we reported the isolation, structural elucidation, and acetylcholinesterase inhibitory activities of **1** and **2**.

## 2. Results and Discussion

Penisimplicin A (**1**, Figure 1) was isolated as a chartreuse oil. The chemical formula of **1** was assigned as C_26_H_29_NO_7_ by the HRESIMS result (*m*/*z* 468.20178 [M + H]^+^, calcd for C_26_H_30_NO_7_, 468.20223), indicating 13 degrees of unsaturation. The ^1^H NMR spectroscopic data of **1** (Table 1) presented five aromatic proton signals at *δ*_H_ 5.97 (1H, s, H-6), 7.38 (1H, d, *J* = 8.0 Hz, H-21), 6.92 (1H, dd, *J* = 8.0, 6.9 Hz, H-22), 7.00 (1H, dd, *J* = 8.1, 6.9 Hz, H-23), and 7.27 (1H, d, *J* = 8.1 Hz, H-24). The ^13^C NMR and DEPT spectroscopic data of **1** (Appendix A) displayed 26 carbon resonances, including 2 methyl carbons, 6 methylene carbons, 7 methines, and 11 quaternary carbons. The coupling patterns and multiplicity of five aromatic proton signals indicated the presence of two benzene rings, a penta-substituted benzene ring (ring A) and 1,2-disubstituted benzene ring (ring B), in the structure of **1**. The HMBC correlations from *δ*_H_ 15.05 (3-OH) to C-2, C-3, C-4, from *δ*_H_ 14.40 (5-OH) to C-4, C-5, C-6, illustrated that there were two hydroxy groups substituted on the *meta* position of benzene ring A (Figure 1). The HMBC correlations from H-6 to C-2, C-4, C-5, C-7, C-8, C-15, and the chemical shift of C-15 (*δ*_C_ 175.5) suggested a carboxyl group attached to C-15 of ring A. An additional six-membered oxygen atom-bearing ring (ring C) fused to ring A was further assigned according to the HMBC correlations from H-1 to C-2, C-3, C-7, C-9, from H-8 to C-2, C-6, C-7, C-9, and the chemical shifts C-1 (*δ*_C_ 66.7), C-9 (*δ*_C_ 67.5). The ^1^H-^1^H COSY correlations of H-9/H_2_-10/H_2_-11/H_2_-12/H_2_-13/H_3_-14 suggested the existence of a pentyl group connected to C-9. The HMBC correlations from *δ*_H_ 10.50 (16-NH) to C-16, C-17, C-18, C-23, from H-22 to C-18, C-20, and from H-19 to C-17, C-21, C-23 allowed the assembly of an indole moiety (including ring B). The above assignments allowed the construction of an alkyl-substituted benzene moiety, which can be designated as a polyketide biosynthetically. In addition, the HMBC correlations from H_2_-24 to C-16, C-17, C-18, and from the methyl protons (*δ*_H_ 3.50) to the carbonyl C-25, suggested the existence of the indole derivative moiety, methyl indoleacetic acetate. Furthermore, the key HMBC correlations from H-1 to C-16, C-17 enabled the connection of the polyketide part with the methyl indoleacetic acetate part by C-1 and C-16 (Figure 2). Hence, the planar structure of **1** was elucidated, as shown in Figure 1.

The absolute configuration of **1** was determined by ECD calculation and DP4+ analysis (Table 2, Figure 3). The ECD calculations of two candidate stereoisomers, **1a** (1*S*,9*R*-**1**) and **1b** (1*S*,9*S*-**1**), were taken into account because the other two stereoisomers, i.e., **1c** (1*R*,9*S*-**1**) and **1d** (1*R*,9*R*-**1**), were enantiomers of **1a** and **1b**. A conformation search of the two stereoisomers at MMFF4s force field was conducted, and the conformers with a distribution higher than 1% were further optimized by density functional theory (DFT) at B3LYP/6-31G(d) level by the Gaussian 16 software package [20] (Appendix A). The obtained stable conformers were subjected to ECD calculation by B3LYP/6-31G(d,p) level of theory. As a result, the ECD calculation results of **1a** and **1b** matched well with the experiment CD spectrum of **1**, which means that the ECD calculations were unable to differentiate between the real and fake stereoisomers in this case. The DP4+ method was a mature strategy with reliable performance in distinguishing stereoisomers of compounds based on the Bayesian analysis of the calculated and experimental NMR data [21]. Therefore, **1a** and **1b** were subjected to NMR calculations and DP4+ analysis. The NMR shielding tensors of two stereoisomers (**1a** and **1b**) were calculated at mPW1PW91/6-31G + (d,p) level of theory. The results were subjected to the DP4+ analysis against the experimental chemical shifts of **1** by the Excel sheet provided by the Sarotti group (https://sarotti-nmr.weebly.com/, accessed on 8 October 2022) [22]. The results of DP4+ analysis showed that the absolute configuration of **1** was 1*S*,9*R* (stereoisomer **1a**, 100% of DP4+ probability) (Table 2). Thus, the structure of **1** was determined and was trivially named as penisimplicin A (Figure 1).

Penisimplicin B (**2**, Figure 1) was obtained as a yellow oil. The molecular formular of **2** was identified as C_25_H_36_N_2_O_7_ by HRESIMS analysis (*m*/*z* 477.25970 [M + H]^+^, calcd for C_25_H_37_N_2_O_7_, 477.26008), suggesting nine degrees of unsaturation. The ^1^H NMR spectroscopic data (Table 1) of **2** presented one aromatic proton singlet *δ*_H_ 5.80 (1H, s, H-6) and three methyl protons signals, at *δ*_H_ 0.86 (3H, t, *J* = 6.9 Hz, H-14), 0.81 (3H, d, *J* = 6.5 Hz, H-25), and 0.80 (3H, d, *J* = 6.4 Hz, H-25). The ^13^C NMR and DEPT spectroscopic data (Table 1) displayed 25 carbon resonances, including 3 methyl carbons, 8 methylene carbons, 6 methines, and 8 quaternary carbons. The data exhibited high similarity to those of compound **1**, implying that compound **2** is a structural congener of **1**. Analysis of the 2D NMR spectra of **2** suggested that the polyketide moiety was nearly the same as that of compound **1**, as evidenced by the HMBC correlations from H-6 to C-2, C-4, C-5, C-7, C-8,C-15, from H-16 to C-2, C-3, C-7, and from H-8 to C-2, C-6, C-7, in combination with the ^1^H-^1^H COSY correlations of H_2_-8/H-9/H_2_-10/H_2_-11/H_2_-12/H_2_-13/H_3_-14 [23]. The remaining 1D NMR signals included two carbonyl groups, two nitrogen atoms, three methylene groups, and three methines, and two methyl doublets. These data were reminiscent of a cyclodipeptide moiety condensed by a proline and leucine or isoleucine. The ^1^H-^1^H COSY correlations of H-16/H_2_-17/H_2_-18/H-19 and of H-21/H-23/H-24, together with the HMBC correlations from H-19 to C-18, C-20, from H-23 to C-21, C-22, C-24, and from H_3_-25 and H_3_-26 to C-23 and C-24, and from *δ*_H_ 7.79 (21-NH) to C-20, C-21, indicated the presence of a cyclic dipeptide consisting of leucine and proline moieties. These assignments were in accordance with the reported data of cyclo(Leu-Pro) [24]. The key HMBC correlations from H-16 to C-2, C-3, and C-7, and from H_2_-17 to C-2, enabled the connection of the polyketide part and cyclodipeptide part by C-2 and C-16. Taken together, the planar structure of **2** was determined, as shown in Figure 2.

The relative configuration of **2** was assigned by the analysis of the ROESY spectrum (Figure 2). The diagnostic ROESY correlation of H-16/H-19/H-21 indicated the co-facial orientations of these three protons. The absolute configurations of the amino acid groups used for assembling the dipeptide part were determined by Marfey’s method [25] (Figure 4). The acid hydrolyzed product of **2** was subjected to make 1-fluro-2,4-dinitrophenyl-5-*L*-alanine amide (FDAA)-derivative under basic conditions. In addition, the FDAA derivatives of the commercially available *D*- and *L*-leucine were also prepared. All the samples were analyzed by LC-MS. As shown in Figure 4, the extracted ion chromatogram (EIC) of D-, *L*-leucine, and **2** suggested that the *L*-leucine FDAA derivative showed the same retention time as the derivative of compound **2** (here, EIC of 384.1514 was used, which corresponds to the exact mass of protonated FDAA-leucine product). Therefore, *L*-leucine was used to construct the cyclopeptide moiety, and absolute configurations of C-16, C-19, and C-21 were determined as *R*, *S*, and *S*, respectively. The remaining absolute configuration of C-9 was determined by computational methods. There are two possible C-9 stereoisomers of **2**, (9*R*,16*R*,19*S*,21*S*)-**2a** and (9*S*,16*R*,19*S*,21*S*)-**2b**, and the real absolute configuration of C-9 can also be differentiated by DP4+ analysis. The results demonstrated that the absolute configuration of **2** was 9*S*,16*R*,19*S*,21*S* (stereoisomer **2b**, 97.91% of DP4+ probability) (Table 3). The absolute configuration of **2** was thus assigned as 9*S*,16*R*,19*S*,21*S*, and named penisimplicin B.

The intriguing structures of these two compounds inspired us to inspect their structural novelty and probable biosynthetic pathways. The alkyl benzene parts of the two compounds are typical polyketides which showed a resemblance to the alkyl benzaldehyde reported in refs. [26,27,28]. The nitrogen-containing moiety of **2** is a cyclodipeptide which is biosynthesized by the cyclodipeptide synthase, whereas for **1**, the nitrogen-containing moiety is most likely biosynthesized from L-tryptophan. Therefore, both compounds **1** and **2** are polyketide–amino acid derivative hybrids, and from this point of view, they are polyketide–nonribosomal peptides in general. However, structurally speaking, these two compounds lack the typical amide bonds which form between the polyketide terminal and the amino group catalyzed by the C or R domains, while each contains a C-C bond between the polyketide and amino acid derivative moieties. This is unprecedented in the PK-NRP family of natural products.

With compounds **1** and **2** in hand, the possible biosynthetic pathways were analyzed and discussed, as shown in Figure 1. The polyketide chain is biosynthesized from an acetyl coenzyme A (CoA) and six molecules of malonyl CoAs, and an *S*-adenosylmethionine, and released as an alkyl benzaldehyde product (A). The methyl group at C-4 of **A** is then oxidized to yield **B** with a carboxylic group. Furthermore, the C-9 carbonyl group of **B** is reduced to yield two stereoisomers, **C** and **D**. Here, the hydroxy group at C-9 is proposed to be formed after the polyketide release, instead of been produced by the KR domain of polyketide synthase. It looks illogical here but is proved to be common in the polyketide biosynthesis [27]. The intermediate C reacts with the *L*-tryptophan derivative, methyl indole acetate, by unknown enzyme(s) to obtain compound **1**. The intermediate D undergoes a cleavage reaction to produce the intermediate E, which further reacts with the cyclo(*L*-leucine-*L*-proline) to yield compound **2**.

Compounds **1** and **2** were subjected to a panel of biological activity assays, including anti-inflammatory activity, but were devoid of any bioactivities. Some nitrogen-containing compounds were reported to show significant acetylcholinesterase inhibitory activity [29,30]. As a result, compound **1** showed significant inhibitory activity against the acetylcholinesterase with IC_50_ 6.35 μM (Appendix A), suggesting that the potential role of this compound be considered as a starting molecule for drug research and development, especially in the field of anti-Alzheimer’s disease drugs. The specific structure-activity relationships need to be further studied due to the great differences of the structures of the two compounds.

## 3. Experimental Section

### 3.1. General Experimental Procedures

Optical rotations were obtained on an Autopol IV-T digital polarimeter (Rudolph, Hackettstown, NJ, USA). UV spectra were recorded on a Hitachi UH5300 spectrophotometer (Hitachi, Tokyo, Japan). CD spectra were measured on a Chirascan Circular Dichroism Spectrometer (Applied Photophysics Limited, Leatherhead, Surrey, UK). One-dimensional and two-dimensional NMR spectra were obtained on a Bruker Avance III 600 MHz spectrometer (Bruker Corporation, Karlsruhe, Germany). HRESIMS data were measured on a Q Exactive Orbitrap mass spectrometer (Thermo Fisher Scientific, Waltham, MA, USA). LC-MS data were measured on an Agilent Q-TOF 6545 system (Agilent Technologies, Santa Clara, CA, USA) equipped with an EC-C_18_ column (particle size 1.9 μm, dimensions 2.1 mm × 50 mm, flow rate 0.2 mL·min^−1^). Preparative high performance liquid chromatography (prep-HPLC) was performed on an Agilent 1260 Infinity II liquid chromatography system equipped with a Zorbax SB-C_18_ column (particle size 5 μm, dimensions 9.4 mm × 150 mm or 7 μm, 21.1 mm × 250 mm, flow rate 5 and 20 mL·min^−1^, respectively) and a DAD detector (Agilent Technologies, Santa Clara, CA, USA). Silica gel (200–300 mesh, Qingdao Haiyang Chemical Co., Ltd., Qingdao, China) and Sephadex LH-20 (GE Healthcare, Uppsala, Sweden) were used for column chromatography (CC). FDAA was bought from Innochem. Acetylcholinesterase (AChE, Sigma-Aldrich, Saint Louis, MO, USA, 0.1 U/mL), acetylthiocholine iodide, dithiobisnitrobenzoic acid (DTNB), and tacrine were bought from Sigma-Aldrich.

### 3.2. Fungal Material

The strain of *Penicillium simplicissimum* JXCC5 was isolated from the root soil of *Ophiocordyceps sinensis* collected from Zhelin Lake Scenic Spot, Jiujiang, Jiangxi Province, in September 2019. A voucher strain (No. JXCC5) was deposited at the Bioactive Natural Products Research Group in South-Central Minzu University, Wuhan, China. The strain of *P. simplicissimum* JXCC5 was activated on glucose and peptone agar (GPA) medium at 25 °C. After seven days, the agar plugs were cut into small pieces to seed twenty 500 mL Erlenmeyer flasks, each containing 200 mL of liquid culture medium (5% glucose, 0.15% peptone from porcine meat, 0.5% yeast extract, 0.05% KH_2_PO_4_, 0.05% MgSO_4_). The flasks were incubated on a rotatory shaker (160 rpm, 25 days) at room temperature in the dark.

### 3.3. Genomic DNA Extraction and Identification

The strain of *Penicillium simplicissimum* JXCC5 was cultured on a glucose–peptone agar medium for 7 days. The mycelia were dried by filter paper and frozen by liquid nitrogen. The cetyltrimethylammonium bromide (CTAB) reagent was then used to extract the genomic sequence [31]. The fungus was identified by the Internal Transcribed Spacer sequence amplified by the primers ITS1 (5′-tccgtgataatcccacttcac-3′) and ITS4 (5′-tcctccgcttattgatatgc-3′). The resulting sequence was submitted to GenBank under accession number OR910619, which showed 100% identification to the recorded entry MN646966.1.

### 3.4. Extraction and Isolation

The content of the culture broth (10 L) of *Penicillium simplicissimum* JXCC5 was centrifuged to separate the mycelium and liquid cultures. The mycelia were soaked with acetone (total 3 L) at room temperature and separated by a centrifuge. The acetone solvent was evaporated to dryness under reduced pressure. The liquid layer was evaporated to a few liters. These two parts of extract were merged and further extracted five times with ethyl acetate (EtOAc) (each time 2 L) to obtain an EtOAc layer (10 L). Afterwards, the EtOAc layer was concentrated in vacuo to obtain 67 g of crude extract. The extract was then fractionated by normal phase silica gel column chromatography (CC) using petroleum ether–acetone mixtures of increasing polarity (20:1, 15:1, 10:1, 5:1, 2:1 to 0:1, *v*/*v*) to yield five fractions (A–E).

Fraction D was subjected to a Sephadex LH-20 CC eluting with MeOH and yielded three subfractions (D1–D3). Subfractions D1 were further separated by normal phase silica gel CC with petroleum ether–acetone mixtures of increasing polarities (*v*/*v*, 15:1, 8:1, 6:1, 4:1, 2:1, 1:1 to 0:1) to yield five subfractions (D1a–D1e).

Subfraction D1b was separated by prep-HPLC (MeCN-H_2_O: 47–57%, 25 min, 4 mL·min^−1^) to afford compound **1** (*t_R_* = 18.40 min, 3.1 mg). Subfraction D1c was separated by prep-HPLC (MeCN-H_2_O: 45–55%, 25 min, 4 mL·min^−1^) to obtain compound **2** (*t_R_* = 19.60 min, 1.9 mg).

### 3.5. Marfey’s Method

Penisimplicin B (0.8 mg) was treated with 300 μL HCl (2 M) at 90 °C under N_2_. After 4 h reaction, the hydrolysate was evaporated and redissolved in 200 μL H_2_O, then 20 μL NaHCO_3_ (1 M) and 20 μL 1-fluoro-2-4-dinitrophenyl-5-L-alanine amide (FDAA, 10 mg/mL in acetone) were added and stirred for 2 h at 45 °C. The reaction was stopped by adding 20 μL HCl (2 M) and was dried in vacuum. The *L*-leucine and *D*-leucine were treated the same way. Finally, the residues were analyzed by LC-MS [MeCN-H_2_O (with 0.5% formic acid): 10:90, 1 min; 10:90-100:0, 1–15 min; 100:0, 15–18 min; 10:90, 18–20 min].

### 3.6. Characterization Data

#### 3.6.1. Penisimplicin A (**1**)

Chartreuse oil; [*α*]^23^_D_ +113.2 (*c* 0.06, MeOH); UV (MeOH) λ_max_ (log *ε*) 230.0 (4.21), 285.0 (3.79) nm; ^1^H NMR (600 MHz, DMSO-*d*_6_) data and ^13^C NMR (150 MHz, DMSO-*d*_6_) data, see Table 1; HRESIMS *m*/*z* 468.20178 [M + H]^+^ (calcd for C_26_H_30_NO_7_, 468.20223).

#### 3.6.2. Penisimplicin B (**2**)

Yellow oil; [*α*]^23^_D_ − 39.3 (*c* 0.06, MeOH); UV (MeOH) λ_max_ (log *ε*) 220.0 (4.08), 250.0 (3.66), 315.0 (3.29) nm; ^1^H NMR (600 MHz, DMSO-*d*_6_) data and ^13^C NMR (150 MHz, DMSO-*d*_6_) data, see Table 1; HRESIMS *m*/*z* 477.25970 [M + H]^+^ (calcd for C_25_H_37_N_2_O_7_, 477.26008).

### 3.7. Computational Details

#### 3.7.1. ECD Calculation

Gaussian 16 package was used for the calculation jobs [20]. The conformers with a distribution higher than 1% when conducted by MMFF94s force field were selected and further optimized by the DFT method at the B3LYP/6-31G(d) level of theory. The theoretical calculation of ECD was performed using time-dependent DFT (TD-DFT) at B3LYP/6-31G(d,p) level in MeOH with the IEFPCM model (Figure 3). The calculated ECD curves were generated according to the Boltzmann weighting of each conformer using SpecDis 1.71 [32] and plotted by an in-house Microsoft Office Excel sheet.

#### 3.7.2. DP4+ Analysis

The conformers optimized at B3LYP/6-31G(d) level were calculated at mPW1PW91/6-31g + (d,p) level in DMSO with the IEFPCM model (Appendix A). The calculated shielding tensors of these conformers were averaged according to the Boltzmann distribution and were applied to a spreadsheet provided by the original publication [21].

### 3.8. Acetylcholinesterase Inhibition Assay

The AChE inhibitory activity was evaluated based on the protocol developed by Ellman et al. [33]. The enzymatic reactions were conducted in 96-well microplates. The following were added to each well: 110 μL phosphate buffer (pH 8.0), 40 μL AChE, and the tested compounds (dissolved in 10 μL DMSO). The microplates were then incubated in 37 °C for 20 min. After that, 40 μL of 1:1 mixture (*v*:*v* = 1:1) of acetylthiocholine iodide (6.25 mM) and DTNB (6.25 mM) were added to each well. Then, the absorbance at 405 nm was detected after 1 h. Seven different concentrations and three replications per concentration for each compound were tested. Tacrine was introduced as a positive control (IC_50_ 0.21 μM). The IC_50_ of compound **1** was calculated by GraphPad Prism 8.0.

### 3.9. Anti-inflammatory Assay

The content of NO was determined by the Griess method [34,35]. Briefly, the murine monocytic RAW264.7 macrophage cells in logarithmic growth phase were diluted to 1 × 105 cells/mL. Then, 100 µL of cell dilution was seeded into the 96-well plates and incubated overnight at 37 °C in a humid atmosphere with 5% CO_2_ for 24 h. The test compound and lipopolysaccharides (LPS) (Sigma) (0.5 µg/mL) were then added to each well and incubated for another 18 h. Then, 50 µL of cells culture supernatant was taken, 50 µL of the Griess reagent (reagent A and reagent B, Sigma) was added in triplicate, and NO production was evaluated in each well. After 5 min of incubation, the absorbances of samples were measured at 570 nm with Envision multilabel plate reader, and the corresponding contents were calculated by curve calibration.

## 4. Conclusions

In this study, two novel nitrogen-containing compounds, penisimplicins A and B, were isolated from the ascomycete *Penicillium simplicissimum* JXCC5. These two compounds are a novel PK-NRP type of compounds and are likely biosynthesized by unprecedented enzymatic machinery. Compound **1** exhibited strong acetylcholinesterase inhibitory activity, which might be a potential candidate for the drug research and development of AD. This study puts the knowledge of the PK-NRP type of natural products one step forward and opens new avenues for the structure types of anti-AD molecules. Further studies on the biosynthetic elucidation and the mode of mechanism of acetylcholinesterase inhibition are needed.

## Data Availability

All data in this research were presented in the manuscript and Appendix A.

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
