# Peer review of "Penisimplicins A and B: Novel Polyketide–Peptide Hybrid Alkaloids from the Fungus Penicillium simplicissimum JXCC5"

_molecules, 2024, doi:10.3390/molecules29030613_

Round 1

Reviewer 1 Report

Comments and Suggestions for Authors

In this manuscript, Ji-Kai Liu and co-workers carried out the chemical investigation on the fungus Penicillium simplicissimum JXCC5, resulting in the isolation and characterization of two novel polyketide-peptide hybrid alkaloids, penisimplicins A (1) and B (2). Their structures were elucidated on the basis of comprehensive spectroscopic data analysis, and their absolute configurations were determined by ECD calculation, DP4+ analysis and Marfey’s method. Moreover, compound 1 exhibited significant inhibitory activity against the acetylcholinesterase with IC50 6.35 μM, suggesting the potential role of this compound been developed as anti-Alzheimer’s drugs. Overall, this work was nice and important.

Only minor revisions were required.

1. In the Introduction section, it is better to give a brief summary of the chemical and biological investigations of the fungus Penicillium simplicissimum.

2. Did you measure the ROESY spectrum? The relative configurations of C-1 and C-9 could be determined by analysis of the ROE correlations, which could help distinguish the real and fake stereoisomers for 1a (1S,9R-1), 1b (1S,9S-1).

Others:

1. Abstract&Keywords: Please use the Italic font for the species name ‘Penicillium simplicissimum’.

2. P1L18: ‘absolute configurations’ → ‘absolute configuration’

3. P1L23: ‘Compound 1’ → ‘compound 1

4. P1L24: ‘IC50 values of 6.35 M’ → ‘IC50 values of 6.35 μM’

5. P2L74: ‘δH 14.40 (5-OH) to C-4’ → ‘from δH 14.40 (5-OH) to C-4’

6. P2L80: ‘H-8 to C-2’ → ‘from H-8 to C-2’

7. P2L89: ‘... HMBC correlation...’ → ‘... HMBC correlations...’

8. P3L100: ‘As a results’ → ‘As a result

9. P3L108: ‘the experimental chemical shifts of 2’ → ‘the experimental chemical shifts of 1

10. Figure 1 and Figure 2 Captions: ‘compounds 1-2’ → ‘compounds 1 and 2

11. Table 1 Caption: Please pay attentions to the formatting of superscripts and subscripts for ‘1H’, ‘13C’and ‘DMSO-d6’.

12. Scheme 1: ‘biosynthesis pathway’ → ‘biosynthesis pathways’

Comments on the Quality of English Language

There are some grammar or typo errors. Some of them are given in the comments to the authors.

Author Response

  1. In the Introduction section, it is better to give a brief summary of the chemical and biological investigations of the fungus Penicillium simplicissimum.

Response: we have added recent research progress of the fungus Penicillium simplicissimum.

  1. Did you measure the ROESY spectrum? The relative configurations of C-1 and C-9 could be determined by analysis of the ROE correlations, which could help distinguish the real and fake stereoisomers for 1a(1S,9R-1), 1b (1S,9S-1).

Response: Yes, we have measured the ROESY spectrum. There is no signal between H-1 and H-9, it could not give a solid evidence to distinguish whether H-1 and H-9 are cofacial or not. However, it also cannot be assigned by analysis of

ECD calculation results. So, the DP4+ analysis was used in this case.

Others:

  1. Abstract&Keywords: Please use the Italic font for the species name ‘Penicillium simplicissimum’.

Response: Revised.

  1. P1L18: ‘absolute configurations’ → ‘absolute configuration’

Response: Revised.

  1. P1L23: ‘Compound 1’ → ‘compound 1

Response: Revised.

  1. P1L24: ‘IC50 values of 6.35 mM’ → ‘IC50values of 6.35 μM’

Response: Revised.

  1. P2L74: ‘δH40 (5-OH) to C-4’ → ‘from δH14.40 (5-OH) to C-4’

Response: Revised.

  1. P2L80: ‘H-8 to C-2’ → ‘from H-8 to C-2’

Response: Revised.

  1. P2L89: ‘... HMBC correlation...’ → ‘... HMBC correlations...’

Response: Revised.

  1. P3L100: ‘As a results’ → ‘As a result’

Response: Revised.

  1. P3L108: ‘the experimental chemical shifts of 2’ → ‘the experimental chemical shifts of 1

Response: Revised.

  1. Figure 1 and Figure 2 Captions: ‘compounds 1-2’ → ‘compounds 1and 2

Response: Revised.

  1. Table 1 Caption: Please pay attentions to the formatting of superscripts and subscripts for ‘1H’, ‘13C’and ‘DMSO-d6’.

Response: Revised.

  1. Scheme 1: ‘biosynthesis pathway’ → ‘biosynthesis pathways’

Response: Revised.

Reviewer 2 Report

Comments and Suggestions for Authors

Based on the reviewer’s expertise in chemical biology and drug discovery, the reviewer suggested rejecting the manuscript in the current format due to significantly insufficient information. However, the authors are encouraged to re-submit the manuscript after major revision to provide more comprehensive information.

The authors provided neither a nucleotide sequence of ITS1 and ITS4 of the claimed fungal species (Penicillium simplicissimum JXCC5) nor deposited the obtained sequence (accession number) on the NCBI database. Therefore, the reviewer cannot evaluate, leading to a severe lack of essential scientific information in the manuscript. Additionally, the provided NCBI recorded entry from the authors did not match the authors’ claim of Penicillium simplicissimum isolate JXCC5. Therefore, more information is required for the reviewer to evaluate.

Furthermore, the authors did not provide information regarding how the author extracted fungal genomic DNA and performed a PCR reaction. This information plays a critical role in fungal DNA barcode identification.

Finally, the authors did not provide sufficient information on the anti-inflammatory and anti-acetylcholinesterase bioactivity claimed in the manuscript. Therefore, the reviewer cannot evaluate the authors’ claim.

For HRESIMS, the entire mass spectra are required for a comprehensive evaluation.

In conclusion, based on the above reasons, the reviewer recommends rejecting this manuscript in its current format due to insufficient scientific information on fungal identification and bioactivities, critical parts of the study.

Comments on the Quality of English Language

-

Author Response

The authors provided neither a nucleotide sequence of ITS1 and ITS4 of the claimed fungal species (Penicillium simplicissimum JXCC5) nor deposited the obtained sequence (accession number) on the NCBI database. Therefore, the reviewer cannot evaluate, leading to a severe lack of essential scientific information in the manuscript. Additionally, the provided NCBI recorded entry from the authors did not match the authors’ claim of Penicillium simplicissimum isolate JXCC5. Therefore, more information is required for the reviewer to evaluate.

Response: the ITS sequence has been submitted to GenBank database under accession number OR910619, this description is also supplemented in section 4.3.

Furthermore, the authors did not provide information regarding how the author extracted fungal genomic DNA and performed a PCR reaction. This information plays a critical role in fungal DNA barcode identification.

Response: we have supplemented the section 4.3 to introduce the procedure of fungal genomic DNA extraction in accordance with the recommendations.

Finally, the authors did not provide sufficient information on the anti-inflammatory and anti-acetylcholinesterase bioactivity claimed in the manuscript. Therefore, the reviewer cannot evaluate the authors’ claim.

Response: we have supplemented the section 4.8 to introduce the procedure of anti-inflammatory bioactivity, and we also supplemented the specific information on the anti-acetylcholinesterase bioactivity as Figure S18.

For HRESIMS, the entire mass spectra are required for a comprehensive evaluation.

Response: we have supplemented the entire LCMS-TIC spectra in supporting information in accordance with the recommendations.

In conclusion, based on the above reasons, the reviewer recommends rejecting this manuscript in its current format due to insufficient scientific information on fungal identification and bioactivities, critical parts of the study.

Reviewer 3 Report

Comments and Suggestions for Authors

p1

Abstract

-accounts for -> account for

-Must be bold, please check the full paper.

penisimplicins A (1) and B (2) -> penisimplicins A (1) and B (2)

-Please keep it italic.

Penicillium simplicissimum -> Penicillium simplicissimum

-Please write in the correct format.

IC50 -> IC50

-It would be better to mark the unit accurately.

6.35 M -> 6.35 µg/mL or 6.35 µM

1. Introduction

-Alzheimer's disease were discovered -> Alzheimer's disease was discovered

-nausea, vomiting, and so on[7] -> nausea, vomiting, and so on[7].

-Providing additional explanations about apolipoprotein E-genotyped patients would be very helpful for better understanding the content.

p2

-It would be better to add several examples to support this sentence.

-Could you unify the units into one at ‘(IC50 = 5.95 µg/mL) from roots of Berchemia floribunda [9], biatractylolide (IC50 = 6.5458 µg/mL) from Atractylodis macrocephalae Rhizoma [10], and isoimperatorin (IC50 = 23.1 µM) and 6'-hydroxy-7'-methoxybergamottin (IC50 = 13.2 µM)’?

-It would be nice to simply mention that botanical gardens are a limited resource in 'However, most of them were isolated from plant sources.'.

2. Results and discussion

-It's more appropriate to write 'Figure S2' rather than 'Table 1'.

The 13C NMR and DEPT spectroscopic data of 1 (Table 1) -> The 13C NMR and DEPT spectroscopic data of 1 (Figure S2)

-It would be better to understand which part is 'ring A' and 'ring B' is drawn in 'Figure 1'.

-It would be better to indicate where the description corresponds (like 'Table 2, Figure 3').

The absolute configuration of 1 was determined by ECD calculation and DP4+ analysis. ->The absolute configuration of 1 was determined by ECD calculation and DP4+ analysis. (Table 2, Figure 3)

p3

-The appearance would be enhanced if the two positions marked as No. 14 in the two structures align in a straight line at Figure 1.

Table 1.

-small letters above the line of text.

1H (600 MHz) and 13C (150 MHz) NMR -> 1H (600 MHz) and 13C (150 MHz) NMR

-Please change it to bold font.

Compounds 1 and 2 -> Compounds 1 and 2

-It would be better to write it in Italic. And it would be better to write down the numbers in subscripts.

(DMSO-d6) -> (DMSO-d6)

-Why is there a line between the numbers "1" and "2"? If there is no need for a special distinction, it would be better to eliminate it.

-better to erase the number of the proton in this table to unite the format.

-Since CH2 is written next to it, it would be a good idea to delete this text in compound 2 No. 8.

p4

-The gap between the table and the sentence is so narrow that it seems stuffy. It's better to widen the gap.

p5

-It would be better to indicate where the description corresponds (like 'Figure 4').

determined by Marfey’s method [22]. -> determined by Marfey’s method [22]. (Figure 4)

-provide information on th types of FDAA, specifically L or D at ‘the FDAA derivatives’.

-the L-leucine FDAA -> the L-leucine FDAA

-Italic, and L or D infront of amino acid should be subscript. Please check it the manuscript at ‘from L-tryptophan.’

from L-tryptophan. -> from L-tryptophan.

p6

- Compound 1 exhibited significant inhibitory activity against acetylcholinesterase with an IC50 of 6.35 μM, while compound 2 showed no efficacy. What factors do you think contribute to this difference at ‘As a result, ~’

-This study put the knowledge -> This study puts the knowledge

p8

-compound 1 (tR = 18.40 min, 3.1 mg). -> compound 1 (tR = 18.40 min, 3.1 mg).

-compound 2 (tR = 19.60 min, 1.9 mg). -> compound 2 (tR = 19.60 min, 1.9 mg).

-It would be better to express the font in bold.

4.5.1. Penisimplicin A (1) -> 4.5.1. Penisimplicin A (1)

p9

-It would be good to understand the description of 'DP4+' in more detail.

p10

-All reference journal names are in italics; Years should be bold.

Author Response

p1

Abstract

-accounts for -> account for

Response: Revised.

-Must be bold, please check the full paper.

penisimplicins A (1) and B (2) -> penisimplicins A (1) and B (2)

Response: Response: Revised.

-Please keep it italic.

Penicillium simplicissimum -> Penicillium simplicissimum

Response: Response: Revised.

-Please write in the correct format.

IC50 -> IC50

Response: Response: Revised.

-It would be better to mark the unit accurately.

6.35 mM -> 6.35 µg/mL or 6.35 µM

Response: the unit has been checked carefully.

  1. Introduction

-Alzheimer's disease were discovered -> Alzheimer's disease was discovered

Response: Revised.

-nausea, vomiting, and so on[7] -> nausea, vomiting, and so on[7].

Response: Revised.

-Providing additional explanations about apolipoprotein E-genotyped patients would be very helpful for better understanding the content.

Response: we have supplemented the explanations about apolipoprotein E-genotyped patients in accordance with the recommendations.

p2

-It would be better to add several examples to support this sentence.

-Could you unify the units into one at ‘(IC50 = 5.95 µg/mL) from roots of Berchemia floribunda [9], biatractylolide (IC50 = 6.5458 µg/mL) from Atractylodis macrocephalae Rhizoma [10], and isoimperatorin (IC50 = 23.1 µM) and 6'-hydroxy-7'-methoxybergamottin (IC50 = 13.2 µM)’?

Response: the unit has been checked carefully and confirmed in accordance with the recommendations.

-It would be nice to simply mention that botanical gardens are a limited resource in 'However, most of them were isolated from plant sources.'.

Response: we have changed the description in accordance with the recommendation.

  1. Results and discussion

-It's more appropriate to write 'Figure S2' rather than 'Table 1'.

The 13C NMR and DEPT spectroscopic data of 1 (Table 1) -> The 13C NMR and DEPT spectroscopic data of 1 (Figure S2)

Response: Revised.

-It would be better to understand which part is 'ring A' and 'ring B' is drawn in 'Figure 1'.

Response: we have drawn the ‘ring A’ and ‘ring B’ in Figure 1 in accordance with the recommendations.

-It would be better to indicate where the description corresponds (like 'Table 2, Figure 3').

The absolute configuration of 1 was determined by ECD calculation and DP4+ analysis. ->The absolute configuration of 1 was determined by ECD calculation and DP4+ analysis. (Table 2, Figure 3)

Response: ‘The absolute configuration of 1 was determined by ECD calculation and DP4+ analysis.’ Has been revised as ‘The absolute configuration of 1 was determined by ECD calculation and DP4+ analysis. (Table 2, Figure 3)’.

p3

-The appearance would be enhanced if the two positions marked as No. 14 in the two structures align in a straight line at Figure 1.

Response: the Figure 1 has been changed in accordance with the recommendations.

Table 1.

-small letters above the line of text.

1H (600 MHz) and 13C (150 MHz) NMR -> 1H (600 MHz) and 13C (150 MHz) NMR

Response: Revised.

-Please change it to bold font.

Compounds 1 and 2 -> Compounds 1 and 2

Response: Revised.

-It would be better to write it in Italic. And it would be better to write down the numbers in subscripts.

(DMSO-d6) -> (DMSO-d6)

Response: Revised.

-Why is there a line between the numbers "1" and "2"? If there is no need for a special distinction, it would be better to eliminate it.

Response: the line has been eliminated in accordance with the recommendations.

-better to erase the number of the proton in this table to unite the format.

Response: the format has been changed in accordance with the recommendations.

-Since CH2 is written next to it, it would be a good idea to delete this text in compound 2 No. 8.

Response: the format has been changed in accordance with the recommendations.

p4

-The gap between the table and the sentence is so narrow that it seems stuffy. It's better to widen the gap.

Response: the gap has been widened in accordance with the recommendations.

p5

-It would be better to indicate where the description corresponds (like 'Figure 4').

determined by Marfey’s method [22]. -> determined by Marfey’s method [22]. (Figure 4)

Response: Revised.

-provide information on th types of FDAA, specifically L or D at ‘the FDAA derivatives’.

-the L-leucine FDAA -> the L-leucine FDAA

Response: Revised.

-Italic, and L or D infront of amino acid should be subscript. Please check it the manuscript at ‘from L-tryptophan.’

from L-tryptophan. -> from L-tryptophan.

Response: Revised.

p6

- Compound 1 exhibited significant inhibitory activity against acetylcholinesterase with an IC50 of 6.35 μM, while compound 2 showed no efficacy. What factors do you think contribute to this difference at ‘As a result, ~’

Response: the structure of the two compounds were in great difference, so we thought the factors contributed to the difference bioactivities needs to be studied by more experiments.

-This study put the knowledge -> This study puts the knowledge

Response: Revised.

p8

-compound 1 (tR = 18.40 min, 3.1 mg). -> compound 1 (tR = 18.40 min, 3.1 mg).

Response: Revised.

-compound 2 (tR = 19.60 min, 1.9 mg). -> compound 2 (tR = 19.60 min, 1.9 mg).

Response: Revised.

-It would be better to express the font in bold.

4.5.1. Penisimplicin A (1) -> 4.5.1. Penisimplicin A (1)

Response: Revised.

p9

-It would be good to understand the description of 'DP4+' in more detail.

Response: we have descripted the ‘DP4+’ in results and discussion section.

p10

-All reference journal names are in italics; Years should be bold.

Response: the formats have been checked and revised in accordance with recommendations.

Reviewer 4 Report

Comments and Suggestions for Authors

The prepared manuscript number "molecules-2724376" "Penisimplicins A and B, novel polyketide-peptide hybrid alkaloids from the fungus Penicillium simplicissimum JXCC5". This study inspects to this work provides a new novel polyketide-peptide hybrid alkaloid.  I have some remarks before they can be published.

1-      The English of the entire manuscript should be significantly revised from a professional to be suitable for publication.

2-      Abstract should be informative and include the main objective.

3-      The introduction section should have more recent data about the Penisimplicins A and B.

4-      At the end of the introduction section, the objective of the work should be more clearly stated.

5-      The data presented hides the amount of work performed. It is noticeable that the work is good and has large results, but the discussion was too descriptive and lacked the explanation of the phenomenon they observed, the authors should discuss the results more deeply with recent references (2020-2023).

6-      The authors mentioned simple application for Compounds 1 and 2, including anti-inflammatory activity and acetylcholinesterase inhibitory activity, but the discussion was too descriptive, lacked the explanation of mode of action.

7-      The results of anti-inflammatory activity assay not existed.  

8-      conclusions should be rewritten not to repeat results but to highlight most important ones and conclude main aspects of the work, implications, and future prospects.

9-      In the reference section the style of some references should be corrected. The Journal names should all either abbreviated or their full names provided.

With kind regards,

Comments on the Quality of English Language

Moderate editing of English language required

Author Response

This study inspects to this work provides a new novel polyketide-peptide hybrid alkaloid. I have some remarks before they can be published.

  • The English of the entire manuscript should be significantly revised from a professional to be suitable for publication.

Response: We have checked the whole manuscript for language errors.

  • Abstract should be informative and include the main objective.

Response: the abstract has been changed in accordance with the recommendations.

  • The introduction section should have more recent data about the Penisimplicins A and B.

Response: the introduction section has been changed in accordance with the recommendations.

  • At the end of the introduction section, the objective of the work should be more clearly stated.

Response: the introduction section has been changed in accordance with the recommendations.

  • The data presented hides the amount of work performed. It is noticeable that the work is good and has large results, but the discussion was too descriptive and lacked the explanation of the phenomenon they observed, the authors should discuss the results more deeply with recent references (2020-2023).

Response: the manuscript has been modified in accordance with the recommendation.

  • The authors mentioned simple application for Compounds 1 and 2, including anti-inflammatory activity and acetylcholinesterase inhibitory activity, but the discussion was too descriptive, lacked the explanation of mode of action.

Response: the results and discussion section has been changed in accordance with the recommendations.

  • The results of anti-inflammatory activity assay not existed.

Response: the anti-inflammatory activity assay has been supplemented as section 4.8.

  • conclusions should be rewritten not to repeat results but to highlight most important ones and conclude main aspects of the work, implications, and future prospects.

Response: the conclusion section has been changed in accordance with the recommendations.

  • In the reference section the style of some references should be corrected. The Journal names should all either abbreviated or their full names provided.

Response: the all formats of references have been checked and revised in accordance with recommendations.

Round 2

Reviewer 2 Report

Comments and Suggestions for Authors

The reviewer appreciates the authors's time and effort in addressing the issue raised by the reviewer. The reviewer has no further issues regarding the current manuscript since the authors can address them all. Therefore, the reviewer is now to recommend the current manuscript to be published in the journal.